# Targeted vibratory therapy as a treatment for proprioceptive dysfunction: Clinical trial in older patients with chronic low back pain

**Yoshihito Sakai**[1]*, **Yoshifumi Morita**[2], **Keitaro Kawai**[2], **Jo Fukuhara**[2], **Tadashi Ito**[3], **Kazunori Yamazaki**[4], **Tsuyoshi Watanabe**[1], **Norimitsu Wakao**[1], **Hiroki Matsui**[1]

1 Department of Orthopaedic Surgery, National Center for Geriatrics and Gerontology, Obu, Aichi Prefecture, Japan, 2 Department of Electrical and Mechanical Engineering, Graduate School of Engineering Nagoya Institute of Technology, Nagoya, Aichi Prefecture, Japan, 3 Three-Dimensional Motion Analysis Room, Aichi Prefectural Mikawa Aoitori Medical and Rehabilitation Center for Developmental Disabilities, Okazaki, Aichi Prefecture, Japan, 4 Institutional Research Center, Aichi Mizuho College, Nagoya, Aichi Prefecture, Japan

* jsakai@ncgg.go.jp

## Abstract

### Introduction

Proprioceptive function declines with age, leading to falls, pain, and difficulties in performing activities of daily living among older adults. Although individuals with low back pain (LBP) exhibit decreased lumbosacral proprioception in various postures, the mechanism by which reduced proprioceptive function causes LBP remains uncertain. Vibratory stimulation may enhance proprioceptive function; however, its efficacy in treating LBP has not been investigated. Thus, we investigated the feasibility of improving proprioceptive function and its effect on alleviating chronic LBP in older patients through targeted vibratory therapy (TVT) administration.

### Methods

This single arm designed trial included older patients aged >65 years with non-specific chronic LBP. TVT involved applying vibratory stimulation, matching the frequency of dysfunctional receptors, for 1 min daily over 14 days to activate proprioceptors; patients performed TVT three times daily at home. In cases of reduced proprioceptive function at multiple sites, TVT was aimed at the lowest frequency band value. LBP and proprioceptive function were evaluated at 2 weeks after TVT and at 2 weeks after the end of TVT in patients with declined proprioception in the trunk or lower extremities.

### Results

Overall, 56 patients with chronic LBP were enrolled; 32 patients were recruited for treatment based on a proprioceptive dysfunction diagnosis and 24 patients were recruited with a normal diagnosis with no significant differences observed between the two sets of patients in sarcopenia-related factors and clinical proprioception-related characteristics. No patient had any adverse events. Two weeks after TVT, the numerical pain rating scale score

**Data Availability Statement:** The data underlying the results presented in the study are available

from Nagoya Institute of Clinical Pharmacology,
http://www.nicp.jp/company/).

**Funding:** This study was supported by the National
Center for Geriatrics and Gerontology in the form
of a grant [21-32] to YS.

**Competing interests:** no competing interests exist.

improved to <3 points in 78.1% of patients, with 73.1% of patients achieving a score of $\leq 3$ points. Proprioceptive function improved in 81.3% of cases, and engagement in activities of daily living improved significantly.

## Conclusions

TVT demonstrated efficacy in improving proprioception and alleviating LBP in older patients with impaired proprioceptive function without affecting non-targeted proprioceptors.

## Introduction

The prevalence of chronic musculoskeletal pain is reported as 15·4% in the Japanese population, with low back pain (LBP) being the most common in men and women aged 75–84 years [1]. One cause of the increasing prevalence of LBP in older people is sarcopenia, an age-related condition that results in skeletal muscle mass reduction [2]. There is no consensus on the causal relationship between trunk muscle atrophy and LBP [3]; however, the fact that LBP is the most common chronic pain occurring in older adults presents a logical discrepancy. Regarding the relationship between skeletal muscles and LBP, the mechanism of segmental stability performed by local muscles, mainly the lumbar multifidus (LM) muscle, has long been recognized as important [4]. The significance of lumbar stability in regulating bipedal walking without LBP explains the rationale for investigating relationship among muscle spindle function, proprioception, and LBP [5]. This perspective highlights the significance of standing stability in the context of postural control and its relevance to LBP. Proprioception, a deep sensory function, detects the position of body parts, the state of muscle movement and contraction, resistance, and weight applied to the body. Proprioceptive feedback influences body movement and positional accuracy even without a visual sense, resulting in associative movements as somatosensory functions in human postural control.

Proprioception encompasses signals from mechanoreceptors found in muscles, tendons, joints, and skin. In addition to muscle spindles in skeletal muscles, Golgi tendon organs located in tendons, Meissner's corpuscles responsible for sensing touch in the skin, and Vater Pacini corpuscles found in the subcutaneous and periosteal tissues are all sensitive to vibration stimuli [6]. Vibratory stimulation of tissues elicits a proprioceptor response, leading to the perception of illusory movement [7]. Muscle-tendon vibration activates afferent nerve fibers, which then transmit signals from the dorsal root of the spinal cord to the cerebrum, resulting in proprioception. Different vibration frequencies are associated with varying degrees of postural control [8].

Although numerous studies on proprioception and LBP have been published [2, 5], the mechanism by which reduced proprioceptive function causes LBP remains unclear. Systematic reviews with meta-analyses have yielded different conclusions regarding the causal relationship between LBP and proprioception. Some studies have suggested a significant relationship between proprioception and LBP [9, 10], while others have reported insufficient evidence [11]. Additionally, there are works indicating a relationship with a small contribution [12], as well as those asserting no significant relationship [13]. Therefore, we developed a diagnostic device to measure proprioceptive function in the trunk and lower limbs that corresponds to the vibration frequencies of all proprioceptors perceivable in humans [14], enabling accurate diagnosis of proprioceptive function. The vibration of skeletal muscles is known to stimulate proprioceptors and activate afferent nerve fibers. Furthermore, vibratory stimulation of the LM muscle in

patients with LBP at a frequency corresponding to that of muscle spindles reportedly improves proprioceptive function [5]. Therefore, to enhance the existing literature on the connection between proprioception and LBP, it is useful to prospectively investigate potential changes in proprioceptive function. Consequently, in this study, we investigated the feasibility of improving proprioceptive function and its effect on alleviating chronic LBP in older patients through the administration of targeted vibratory therapy (TVT).

## Methods

### Study design and participants

Patients with non-specific chronic LBP were recruited from a single institute (National Center for Geriatrics and Gerontology) during the period from April 2021 to March 2023. Participants were briefed on the study and each of them provided informed consent. Eligible participants were older people aged > 65 years with LBP without sciatica for at least 6 months. The participant flow diagram shows the progress of the study subject in Fig 1. We limited the study to patients aged > 65 years because postural responses to proprio-muscular inputs are known

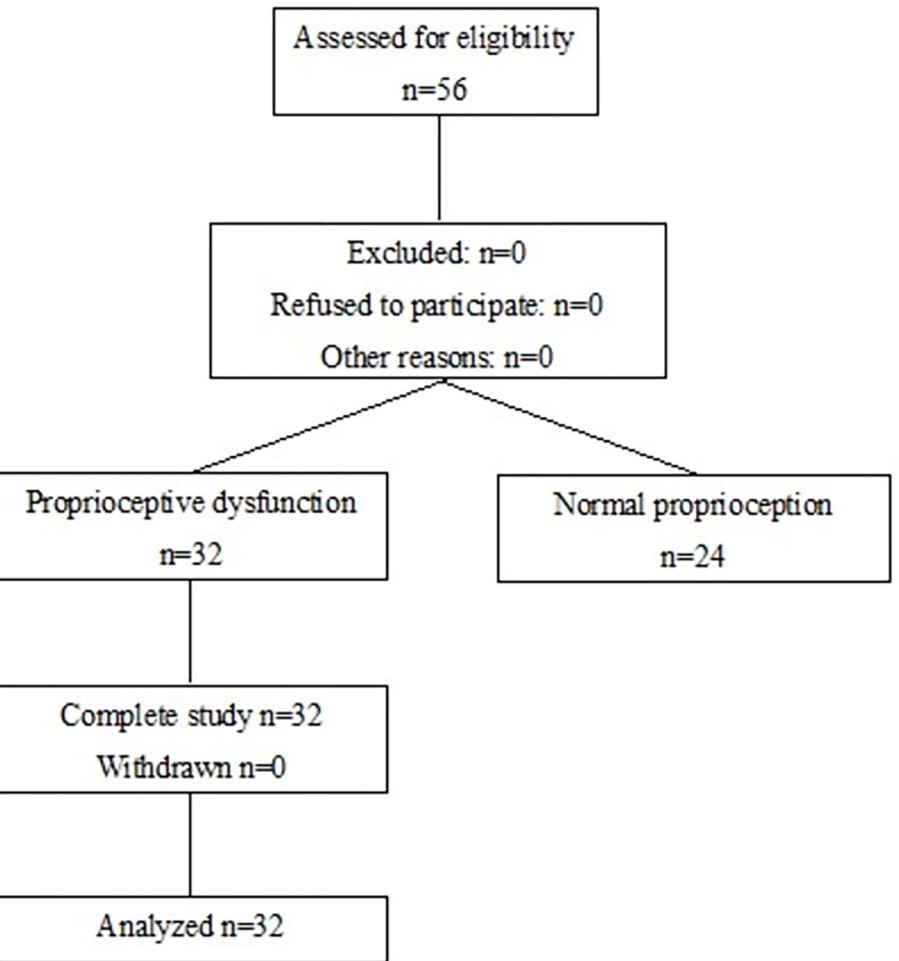

**Fig 1. Participant flow diagram.** Fifty-six patients with chronic low back pain were enrolled, and all patients underwent proprioceptive function diagnostic evaluation. Among them, 32 patients were diagnosed with proprioceptive dysfunction, and all 32 patients were treated with TVT for 2 weeks.

to be affected by aging, particularly in individuals aged $> 60$ years [15]. In this study, non-specific LBP was defined as follows: (1) LBP with numerical pain rating scale (NRS) scores $\geq 3$ points [16]; (2) persistent pain localized below the costal margin and above the inferior gluteal folds; (3) absence of specific spinal pathologies, such as infection, tumors, and vertebral fractures on both plain radiographs and lumbar magnetic resonance imaging (MRI); (4) no dominant leg pain caused by radicular and cauda equina disorders; (5) no significant instability, such as spondylolysis, isthmic spondylolisthesis, and degenerative spondylolisthesis more than grade II; and (6) no previous lumbar and thoracolumbar spine surgery. All patients underwent Fall Risk Index evaluation [17], 10-m walking speed measurement with normal gait, grip strength measurement (left and right average), five times chair rising test, whole-spine standing radiography, lumbar spine MRI, and dual-energy X-ray absorptiometry (DXA). Spinal sagittal alignment was evaluated by measuring lumbar lordosis (LL) from L1 to S1, lumbar range of motion (L-ROM) from L1 to S1, sagittal vertical axis (SVA), thoracic kyphosis angle (TK), pelvic tilt (PT), pelvic incidence (PI), and PI minus lumbar lordosis (PI-LL) in lateral view. MRI T2-weighted axial images were obtained to measure the cross-sectional area of the back muscles (sum of the left and right LM and erector spinae muscles) at the L4/5-disc level. Bone mineral density (T-score) at the L2–4 level and the skeletal muscle mass index (SMI) (kg/m$^2$), calculated by dividing the skeletal muscle mass of the extremities by the square of height, were obtained using DXA. All patients were enrolled after confirming that their persistent pain could not be relieved using nonsteroidal anti-inflammatory drugs (NSAIDs) and opioid therapy (tramadol) for $>1$ month. We excluded patients who could not stand unsupported, scored $>6$ points on the Fall Risk Index [17], or could not be evaluated using NRS owing to dementia.

This study was conducted according to the Declaration of Helsinki guidelines, and ethical approval was obtained from the University of Fujita Health University Certified Review Board (CRB4180003). It was registered with the jRCT (jRCTs042200058) prior to participant enrollment, and all participants provided written informed consent.

## Procedure

**Diagnosis of proprioception.** The diagnostic equipment developed by our research team was employed to assess proprioceptive function (Fig 2). This device administers continuous vibrations ranging from 27 to 272 Hz separately to the trunk and lower legs in a sweeping manner. It gauges the biological response to the vibratory stimuli as an electrical signal indicating the displacement of the center of pressure (CoP) using a stabilometer (T.K.K.5810, TAKEI Scientific Instruments Co., Ltd., Niigata, Japan) while the patients stand with their eyes closed. Vibratory stimuli, consisting of four vibrators (NSW1-205-8A, 5 W, 8 Ω, Aurasound, Inc., Santa Fe Springs, CA, USA), were applied to the bilateral gastrocnemius-soleus (GS) and LM. The vibration section was divided into three evaluation section intervals of 15 seconds each (ES$_{1,2,3}$) determined according to the frequency of the vibratory stimulation corresponding to the proprioceptors. The subscript numbers of ES$_{1,2,3}$ are indicated as follows: 1. muscle spindles (lower frequency), from 30–53 Hz; 2. muscle spindles (higher frequency), from 56–100 Hz; and 3. Vater–Pacini corpuscle, from 140–250 Hz [18]. The sites where vibratory stimulation was applied were the GS and LM muscles, with each measurement time of 75 s consisting of the first 15 s without vibration (pre-section) and 60 s with vibration (vib-section). The CoP was measured by applying local vibratory stimulation to the GS or LM muscles, with the patients' eyes closed. The root mean square (RMS) amplitude during the vibration of the GS and LM muscles was used as an indicator to evaluate the magnitude of the CoP in the anterior–posterior direction [19]. In this study, the modified RMS was calculated by determining

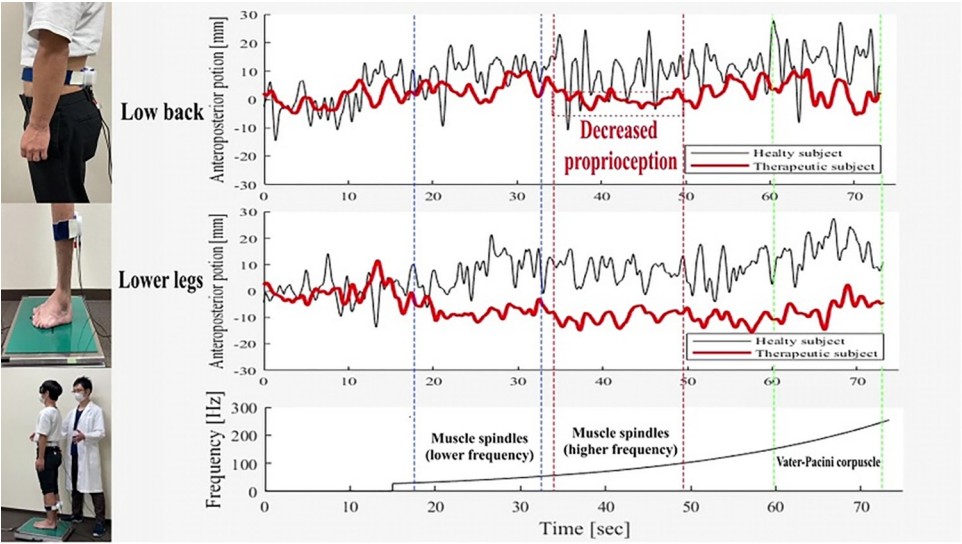

**Fig 2. Diagnosis of proprioceptive function in the trunk and lower extremity.** Vibratory stimulation covering a wide range of frequencies from 27 Hz to 272 Hz targets nearly all proprioceptors and is administered for diagnosing proprioceptive functions in older adults. The CoP displacement was continuously analyzed as a waveform using a stabilometer. Vibrations were applied to the bilateral lumbar multifidus and gastrocnemius soleus muscles, and mechanical vibrations from low to high frequencies were applied continuously. The displacement of the COP during vibration at the trunk and calf was automatically recorded using the stabilometer to evaluate proprioceptor function.

the difference in CoP data between the pre- and vib- sections as follows.

$$\text{RMS}^1_* = \frac{\sqrt{\frac{1}{N}\sum_{n=n_1}^{n_2}\left\{Y_{\text{Vib}(*)}(n) - \bar{Y}_{\text{pre}(*)}\right\}^2}}{\text{RMS}^{\text{pre}}_*}$$

$$\text{RMS}^2_* = \frac{\sqrt{\frac{1}{N}\sum_{n=n_3}^{n_4}\left\{Y_{\text{Vib}(*)}(n) - \bar{Y}_{\text{pre}(*)}\right\}^2}}{\text{RMS}^{\text{pre}}_*}$$

$$\text{RMS}^3_* = \frac{\sqrt{\frac{1}{N}\sum_{n=n_5}^{n_6}\left\{Y_{\text{Vib}(*)}(n) - \bar{Y}_{\text{pre}(*)}\right\}^2}}{\text{RMS}^{\text{pre}}_*}$$

The number N was equal to 300 because of the 15-s ES intervals, and the sampling frequency was 20 Hz. The RMSs were evaluated as the biological response to vibratory stimulation, indicating a proprioceptive function relative to the RMS value obtained without vibration. The greater the sway in each ES in the vibration section, the greater the RMS. A better proprioceptive function corresponds to a greater transition in the anteroposterior direction during vibratory application. Therefore, a higher RMS value indicates a superior proprioceptive function.

For diagnosis of reduced proprioceptive function, RMS data from 81 young adults were used to identify the site and proprioceptors. An RMS value of -1 standard deviation (SD) relative to young adults was defined as proprioceptive dysfunction [19]. The RMS cutoff values for

each site and frequency band were as follows.

$$\mathrm{RMS}_{\mathrm{GS}}^{1}\ (30-53\ \mathrm{Hz}):\ 1.01$$

$$\mathrm{RMS}_{\mathrm{GS}}^{2}\ (56-100\ \mathrm{Hz}):\ 1.02$$

$$\mathrm{RMS}_{\mathrm{GS}}^{3}\ (140-250\ \mathrm{Hz}):\ 0.69$$

$$\mathrm{RMS}_{\mathrm{LM}}^{1}\ (30-53\ \mathrm{Hz}):\ 0.85$$

$$\mathrm{RMS}_{\mathrm{LM}}^{2}\ (56-100\ \mathrm{Hz}):\ 0.84$$

$$\mathrm{RMS}_{\mathrm{LM}}^{3}\ (140-250\ \mathrm{Hz}):\ 0.65$$

**TVT for proprioceptive dysfunction.** TVT was performed on the portion of reduced function where the RMS was lower than the cutoff value. This was achieved by applying vibratory stimulation corresponding to the frequency of dysfunctional receptors for 1 min to activate the proprioceptors. The frequencies for treatment output were calculated based on a previous study [14]. The vibratory intervention programs of this trial are structured according to previously published guidelines for whole-body vibration studies in information about the device and vibration [20]. The device used in this therapeutic trial is non-commercial (self-built) and uses the same vibrators as the diagnostic device described above. Briefly, the device is 60 mm * 75 mm * 3 mm large, has a mass of 52.3 g, and can deliver sinusoidal (deviation <5%) vibration per vibrator. The device can produce vibrations with peak-to-peak displacement up to 1.6 mm within a frequency range of 30–53Hz, 56–00Hz, and 140–250Hz. The vibrator produced sinusoidal accelerations primarily in the vertical component, with minimal variation in acceleration direction throughout each sinusoidal cycle.

Therapeutic vibratory stimulation lasted 60 s in each frequency band; the frequency rising and descending patterns were continuous for 30 s in the frequency band. The equipment was loaned to patients with proprioceptive dysfunction for use in either the trunk or lower extremities, covering both frequency bands; it was used for 14 days, with patients performing TVT three times daily at home. In cases where proprioceptive function was impaired across multiple sites, the targeted treatment was aimed at the frequency band exhibiting the lowest RMS value. Patients were instructed to maintain a daily treatment diary, and any instances of missing treatments exceeding 10% were excluded from the analysis. Based on a previous study indicating that proprioceptive function reaches saturation within 1 min of vibratory stimulation [21], we set a treatment duration of 1 min per session for TVT. Regarding the treatment periods, a previous study administering opioids for chronic LBP demonstrated significant pain improvement after 2 weeks of medication administration [22]. As we anticipated a treatment effect equivalent to pharmacotherapy for chronic LBP, we set the treatment period at 2 weeks. The clinical evaluation periods were pre-intervention, 2 weeks post-intervention, and 2 weeks following the conclusion of the intervention.

**Outcome assessment.** Changes in the intensity of LBP, disability caused by LBP, quality of life, mental quality, and central sensitization were assessed before the intervention and at the 2- and 4-week follow-up examinations. Pain intensity was measured on an NRS and scored from 0 (no pain) to 10 (worst possible pain) points. Disability resulting from LBP was evaluated using the Roland–Morris Disability Questionnaire (RDQ), with scores ranging from 0 (no disability) to 24 (highest disability) points [23]. Quality of life was assessed using the

EuroQol-5D (EQ-5D) and EuroQol-visual analog scale. The EQ-5D provides a utility score anchored at 0 for death and 1 for perfect health, while the EuroQol-visual analog scale scores range from 0 for the worst health to 100 points for the best health [24]. The Geriatric Depression Scale-Short Form was used to assess depression among older adults. It is a 15-item yes–no self-report measure with a maximum score of 15 points, where higher scores represent more severe symptoms [25]. Central sensitization, an abnormal heightened sensitivity arising when the functionality of nociceptive pathways in the central nervous system is enhanced, was assessed using Central Sensitization Inventory (CSI) with central sensitization-related items with total scores ranging from 0 to 100 points [26].

**Data management and statistical analysis.** The aim was to demonstrate the same clinical difference in the NRS scores as those achieved following 2-week opioid therapy for older patients with chronic LBP [22]. This suggested a difference in the NRS of 1.7 (SD, 3.1) points. A paired t-test with a two-sided significance level of 0.05 and 80% power requires a sample size of 29 patients to be recruited.

We compared the baseline characteristics of patients with and without proprioceptive dysfunction using the Mann–Whitney U test. Improvements in LBP, activities of daily living (ADLs), and proprioceptive function were assessed by measuring changes before and after treatment. The statistical significance of the differences in pain intensity, ADLs, and proprioceptive function was compared between pre-treatment and 2 weeks after treatment, 2 weeks and 4 weeks after treatment, and pre-treatment and 4 weeks after treatment using the Wilcoxon signed-rank test. The results are presented as mean differences for continuous outcomes with 99% confidence intervals (CIs).

All statistical analyses were performed using the EZR software (Saitama Medical Center, Jichi Medical University, Saitama, Japan). The level of significance was set at $p$-value $<0.01$.

## Results

Ultimately, 56 patients with chronic LBP (mean age, 76.38±5.30 years; minimum, 67 years; median, 76 years; maximum, 87 years; sex, 28 male and 28 female individuals) were enrolled, and all patients underwent proprioceptive function diagnostic evaluation. Among them, 32 patients were diagnosed with proprioceptive dysfunction in either the trunk or lower extremities, with RMS values below the cutoff value, which is the standard for young, healthy participants. All 32 patients with chronic LBP were treated with TVT for 2 weeks to reduce proprioceptive dysfunction. The proportions and frequency bands of proprioceptive dysfunction were as follows: lumbar spine, 17 cases; lower limbs, 15 cases; 30–53 Hz, 10 cases; 56–100 Hz, 19 cases; and 140–250 Hz, three cases. Six patients had decreased proprioceptive function at multiple sites and frequency bands. No adverse events were observed in any patient. All patients completed 2 weeks of intervention and underwent follow-up evaluation for 2 weeks after treatment completion.

A comparison between the 32 patients with impaired proprioceptive function and 24 patients with normal function is presented in Table 1. No significant differences were observed between the two groups in factors related to sarcopenia, such as grip strength, gait speed, and SMI. Furthermore, no differences were observed in clinical characteristics, including body composition, center-of-gravity test results without vibratory stimulation, and radiographic findings.

During the 2-week TVT and 2-week post-intervention periods, 25 of 32 (78.1%) patients experienced an NRS score improvement to <3 points with intervention, and 11 (45.8%) of these patients experienced a worsened NRS score of ≥3 points at 2 weeks after treatment completion. The mean NRS score after the 2-week TVT demonstrated significant improvement

**Table 1. Clinical characteristics in older patients with chronic low back pain with or without proprioceptive dysfunction.**

| | Normal function N = 24 | Proprioceptive dysfunction N = 32 | p value |
|---|---|---|---|
| RMS | 1.28 (0.34) | 0.77 (0.15) | <0.0001 |
| Age (year) | 76.68 (5.80) | 76.16 (5.00) | 0.73 |
| Sex (male: female) | 13:11 | 15:17 | 0.79 |
| BMI (kg/m$^2$) | 23.73 (4.20) | 24.57 (3.60) | 0.43 |
| Affected period of LBP (months) | 91.71 (48.76) | 100.00 (59.64) | 0.58 |
| NRS | 6.48 (1.33) | 6.31 (1.28) | 0.62 |
| RDQ | 9.94 (3.16) | 8.79 (3.90) | 0.23 |
| EQ-5D | 0.66 (0.18) | 0.62 (0.23) | 0.43 |
| EQ-VAS | 64.25 (15.60) | 62.50 (14.74) | 0.67 |
| GDS | 2.88 (3.22) | 3.88 (3.01) | 0.24 |
| CSI | 15.34 (9.71) | 17.50 (10.10) | 0.42 |
| Fall risk index | 1.17 (1.55) | 1.50 (1.69) | 0.45 |
| 10 m walk speed (m/sec) | 1.10 (0.18) | 1.09 (0.18) | 0.86 |
| Grip strength (kg) | 25.33 (7.47) | 25.80 (7.69) | 0.82 |
| Chair rising test (sec) | 10.75 (4.93) | 10.05 (2.90) | 0.51 |
| Total locus length (eye open) (mm) | 415 (125) | 363 (105) | 0.092 |
| Total locus length (eye closed) (mm) | 603 (232) | 544 (189) | 0.30 |
| Outer peripheral area (eye open) (mm$^2$) | 412 (225) | 321 (167) | 0.087 |
| Outer peripheral area (eye closed) (mm$^2$) | 963 (775) | 701 (482) | 0.13 |
| Romberg ratio (length) | 1.45 (0.37) | 1.49 (0.28) | 0.57 |
| Romberg ratio (area) | 2.57 (1.90) | 2.18 (1.11) | 0.35 |
| Antero-posterior RMS (eye open) | 5.13 (1.80) | 4.46 (1.43) | 0.12 |
| Antero-posterior RMS (eye closed) | 6.93 (2.67) | 6.50 (3.11) | 0.59 |
| Bone mineral density (L2-4 T-score) | 0.21 (2.22) | 0.50 (2.53) | 0.66 |
| SMI (kg/m$^2$) | 6.61 (0.99) | 6.67 (1.03) | 0.82 |
| Body fat ratio (%) | 31.91 (6.62) | 33.07 (8.0) | 0.57 |
| Trunk muscle CSA | 1790 (581) | 1554 (473) | 0.010 |
| LL (degree) | 29.92 (11.17) | 25.00 (13.56) | 0.15 |
| L-ROM (degree) | 28.87 (13.47) | 26.31 (11.78) | 0.46 |
| SVA (mm) | 69.96 (42.01) | 72.07 (38.29) | 0.85 |
| TK (degree) | 38.12 (10.09) | 37.00 (11.61) | 0.71 |
| PT (degree) | 26.92 (10.21) | 26.47 (8.79) | 0.86 |
| PI-LL (degree) | 20.75 (12.63) | 19.91 (13.90) | 0.82 |

Data are presented as means (SDs). The *p*-values are indicated to two significant digits.

The root mean square (RMS) was calculated by the difference in CoP data between the pre- and vib- sections. (See Diagnosis of proprioception).

BMI: body mass index, LBP: low back pain, NRS: numerical pain rating scale, RDQ: Roland-Morris Disability Questionnaire, EQ-5D: EuroQOL five dimensions questionnaire, EQ-VAS: EuroQOL visual analogue scale, GDS: Geriatric Depression Scale-Short Form, CSI: Central Sensitization Inventor, Total locus length: the sum of the path length of center of pressure, Outer peripheral area: the area that encloses the circumference of the stabilogram, SMI: skeletal muscle index, CSA: cross sectional area, LL: lumbar lordosis, L-ROM: lumbar ROM, SVA: sagittal vertical axis, TK: thoracic kyphosis, PT: pelvic tilt, PI-LL: pelvic incidence minus lumbar lordosis

than that before the intervention (*p*<0.0001). However, relative to the score at the end of the intervention, the mean NRS score at 2 weeks after the end of treatment increased from 2.6 to 3.6 points; nevertheless, this difference was not statistically significant (*p* = 0.0162).

Significant improvements in the RDQ, EQ-5D, and CSI assessments were observed after the 2-week TVT. No significant differences in ADL parameters were found between the intervention's conclusion and 2 weeks after its completion (Table 2). The results of the pre- and

**Table 2. Results of targeted vibratory therapy for proprioceptive dysfunction.**

|  | Before treatment | 2 weeks (treatment+) | 4 weeks (treatment -) | Mean difference 99% CI p value [1] | Mean difference 99% CI p value [2] | Mean difference 99% CI p value [3] |
|---|---|---|---|---|---|---|
| **NRS** | 6.31 (1.28) | 2.61 (1.47) | 3.61 (2.20) | -3.69 -4.49 to -2.90 <0.0001 | 1.00 0.11 to 1.88 0.016 | -2.70 -3.65 to -1.74 <0.0001 |
| **RDQ** | 9.94 (3.16) | 4.50 (4.21) | 4.22 (4.07) | -5.44 -6.69 to -4.19 <0.0001 | -0.28 -1.33 to 0.77 0.58 | -5.72 -7.23 to -4.20 <0.0001 |
| **EQ-5D** | 0.66 (0.18) | 0.83 (0.15) | 0.85 (0.17) | 0.17 0.09 to 0.25 <0.0001 | 0.02 -0.04 to 0.08 0.50 | 0.18 0.11 to 0.26 <0.0001 |
| **EQ-VAS** | 64.25 (15.60) | 71.56 (15.70) | 67.16 (16.33) | 7.31 0.69 to 13.93 0.0057 | -4.41 -9.14 to 0.33 0.020 | 2.91 -4.79 to 10.60 0.30 |
| **GDS** | 2.88 (3.22) | 2.13 (3.20) | 1.69 (3.02) | -0.75 -1.59 to 0.09 0.017 | -0.44 -1.10 to 0.23 0.091 | -1.19 -2.34 to -0.04 0.0048 |
| **CSI** | 15.34 (9.71) | 10.38 (6.75) | 8.53 (6.99) | -4.97 -7.85 to -2.09 <0.0001 | -1.84 -4.20 to 0.51 0.050 | -6.81 -11.11 to -2.52 0.00020 |

Data are presented as means (SDs). The *p*-values are indicated to two significant digits.

Treatment+ indicates 2 weeks after treatment and treatment- indicates 2 weeks after the end of treatment

*p* value [1]: 2 weeks after treatment vs before treatment, *p* value [2]: 4 weeks vs 2 weeks after treatment, *p* value [3]: 4 weeks vs before treatment according to a Wilcoxon signed-rank test.

NRS: numerical pain rating scale for low back pain, RDQ: Roland-Morris Disability Questionnaire, EQ-5D: EuroQOL five dimensions questionnaire, EQ-VAS: EuroQOL visual analogue scale, GDS: Geriatric Depression Scale-Short Form, CSI: Central Sensitization Inventory

post-intervention center-of-gravity tests and RMS values are presented in Tables 3 and 4, respectively. The results of the ordinary center-of-gravity test revealed no significant changes after the intervention or at the end of treatment. However, the mean RMS value of the TVT site, where the RMS value in each frequency band was below the cutoff value, demonstrated a significant improvement, increasing from a mean of 0.774 before treatment to 1.709 after 2 weeks of TVT ($p<0.0001$). Subsequently, the RMS value decreased to 1.567 at 2 weeks after the end of treatment, with no significant difference from the value observed after 2 weeks of intervention ($p = 0.901$). Among the 32 patients, 26 (81.3%) demonstrated improved RMS values above the cutoff values after 2 weeks of TVT and 19 (73.1%) achieved an NRS score of ≤3 points.

## Discussion

To our knowledge, this is the first prospective study to evaluate the efficacy of TVT as a treatment for proprioceptive function, and it represents the first clinical study to establish a medical opinion on the relationship between proprioceptive function and LBP derived from an intervention study in patients with nonspecific chronic LBP. A unique aspect of this study is the evaluation of LBP improvement in older patients with impaired proprioceptive function, using an objective diagnosis of proprioceptive decline to target the impaired proprioceptors. Methods for activating proprioceptors through vibratory stimulation to achieve proprioceptive improvement have been reported previously [5, 21]; however, this is the first attempt to use a therapeutic method for identifying receptors and sites with impaired function. The therapeutic vibratory approach, which targets specific proprioceptors with impaired function, effectively

**Table 3. Targeted vibratory therapy-related changes in the swaying center of gravity without vibration.**

| | Before treatment | 2 weeks (treatment+) | 4 weeks (treatment -) | Mean difference 99% CI p value [1] | Mean difference 99% CI p value [2] | Mean difference 99% CI p value [3] |
|---|---|---|---|---|---|---|
| **Total locus length (eye open) (mm)** | 362.68 (104.70) | 366.21 (121.62) | 361.54 (120.66) | 3.53 -63.67 to 70.73 0.96 | -4.66 -35.67 to 26.35 0.37 | -1.13 -69.13 to 66.87 0.96 |
| **Total locus length (eye closed) (mm)** | 544.39 (188.60) | 519.12 (188.16) | 495.18 (190.97) | -25.27 -139.75 to 89.22 0.35 | -23.95 -66.11 to 18.22 0.13 | -49.21 -160.49 to 62.07 0.13 |
| **Outer peripheral area (eye open) (mm$^2$)** | 321.41 (166.56) | 352.44 (245.94) | 376.02 (296.45) | 31.02 -98.86 to 160.91 0.87 | 23.58 -71.88 to 119.04 0.82 | 54.61 -108.70 to 217.91 0.57 |
| **Outer peripheral area (eye closed) (mm$^2$)** | 700.69 (482.10) | 681.30 (418.30) | 626.49 (473.08) | -19.39 -295.67 to 256.89 0.36 | -54.81 -224.62 to 114.00 0.18 | -74.20 -348.70 to 200.29 0.36 |
| **Romberg ratio (length)** | 1.49 (0.28) | 1.43 (0.30) | 1.37 (0.29) | -0.07 -0.28 to 0.14 0.098 | -0.05 -0.20 to 0.09 0.38 | -0.12 -0.32 to 0.08 0.11 |
| **Romberg ratio (area)** | 2.18 (1.11) | 2.09 (0.86) | 1.82 (0.99) | -0.09 -0.84 to 0.65 0.93 | -0.27 -0.87 to 0.33 0.21 | -0.36 -1.09 to 0.37 0.37 |
| **Antero-posterior RMS (eye open)** | 4.46 (1.43) | 4.96 (1.84) | 5.00 (2.02) | 0.50 -0.40 to 1.40 0.16 | 0.04 -0.72 to 0.81 0.85 | 0.55 -0.67 to 1.76 0.30 |
| **Antero-posterior RMS (eye closed)** | 6.50 (3.11) | 6.31 (2.24) | 6.03 (2.36) | -0.20 -2.00 to 1.61 0.76 | -0.28 -1.27 to 0.71 0.54 | -0.48 -1.96 to 1.00 0.47 |

Data are presented as means (SDs). The *p*-values are indicated to two significant digits.

Treatment+ indicates 2 weeks after treatment and treatment- indicates 2 weeks after the end of treatment

The root mean square (RMS) was calculated by the difference in CoP data between pre- and vib- sections. (See Diagnosis of proprioception)

The analysis of CoP was performed using the following parameters: AP and ML range, sway velocity, length (cm), rectangle-area (cm$^2$), Romberg's quotient of length (%), and Romberg's quotient of rectangle-area (%).

*p* value [1]: 2 weeks after treatment vs before treatment, *p* value [2]: 4 weeks vs 2 weeks after treatment, *p* value [3]: 4 weeks vs before treatment according to a Wilcoxon signed-rank test.

CoP: center of pressure, AP: antero-posterior, ML: medio-lateral

improves proprioception and prevents proprioceptive deterioration caused by harmful vibratory stimulation in healthy proprioceptors [27]. Therefore, it is essential to identify the specific proprioceptors and sites of impaired function before intervention.

Among older patients with chronic LBP, 57.1% exhibited reduced proprioceptive function, with dysfunction sites almost equally distributed between the trunk and lower extremities. This finding supports the research conclusions that chronic stress loading, arising from the overactivity of trunk proprioceptive function, can induce pain [28]. This occurs when the proprioceptive function of the lower limbs is reduced in addition to the trunk in older patients with chronic LBP [27]. Notably, patients diagnosed with proprioceptive dysfunction in chronic LBP displayed no distinctive clinical findings compared to those with normal proprioceptive function. This includes factors such as body composition or center-of-gravity sway, indicating that vibratory stimulation is essential for detecting proprioceptive pain.

All patients with chronic LBP and impaired proprioceptive function who underwent TVT experienced long-term pain and did not respond to 1 month of drug administration, with no

**Table 4. Targeted vibratory therapy-related changes in proprioceptive function (RMS).**

| | Before treatment | 2 weeks (treatment+) | 4 weeks (treatment -) | Mean difference 99% CI p value [1] | Mean difference 99% CI p value [2] | Mean difference 99% CI p value [3] |
|---|---|---|---|---|---|---|
| **Lower limb 30-53Hz** | 1.500 (0.731) | 1.735 (0.879) | 1.557 (0.591) | 0.24 -0.29 to 0.76 0.26 | -0.17 -0.65 to 0.30 0.55 | 0.06 -0.29 to 0.40 0.54 |
| **Lower limb 56-100Hz** | 1.449 (0.792) | 1.851 (1.046) | 1.699 (1.034) | 0.40 -0.20 to 1.01 0.13 | -0.15 -0.75 to 0.45 0.34 | 0.25 -0.31 to 0.81 0.19 |
| **Lower limb 150-250Hz** | 1.350 (0.689) | 1.769 (0.830) | 1.657 (0.686) | 0.42 0.01 to 0.82 0.0099 | -0.11 -0.50 to 0.28 0.97 | 0.31 -0.09 to 0.70 0.049 |
| **Lumbar spine 30-53Hz** | 1.422 (0.710) | 1.502 (0.680) | 1.713 (1.156) | 0.08 -0.41 to 0.57 0.54 | 0.21 -0.43 to 0.85 0.60 | 0.29 -0.25 to 0.84 0.27 |
| **Lumbar spine 56-100Hz** | 1.388 (0.709) | 1.745 (0.789) | 1.797 (1.106) | 0.36 -0.09 to 0.80 0.042 | 0.05 -0.62 to 0.72 0.40 | 0.40 -0.21 to 1.02 0.082 |
| **Lumbar spine 150-250Hz** | 1.412 (0.754) | 1.608 (0.830) | 1.549 (0.778) | 0.20 -0.17 to 0.57 0.12 | -0.06 -0.53 to 0.42 0.44 | 0.14 -0.31 to 0.58 0.73 |
| **Most affected RMS** | 0.774 (0.153) | 1.709 (0.838) | 1.567 (0.709) | 0.94 0.52 to 1.35 <0.0001 | -0.14 -0.57 to 0.29 0.42 | 0.79 0.44 to 1.15 <0.0001 |

Data are presented as means (SDs). The *p*-values are indicated to two significant digits.

Treatment+ indicates 2 weeks after treatment and treatment- indicates 2 weeks after the end of treatment

The root mean square (RMS) was calculated by the difference in CoP data between pre- and vib- sections. (See Diagnosis of proprioception)

*p* value [1]: 2 weeks after treatment vs before treatment, *p* value [2]: 4 weeks vs 2 weeks after treatment, *p* value [3]: 4 weeks vs before treatment according to a Wilcoxon signed-rank test.

anticipated carry-over effects or pain relief owing to natural history. The observation that a 2-week course of TVT resulted in pain relief of 3.7 points (60%) on NRS, accompanied by notable enhancements in ADLs in line with subjective pain assessment, is considered a favorable outcome associated with TVT. Among the 81.3% of older patients with chronic LBP whose proprioceptive function recovered to the level of younger healthy individuals, 73.1% had improved LBP with an NRS score of ≤3 points. Overall, 59.4% (19 of 32) of patients demonstrated improvements both in proprioceptive function and LBP after TVT. After treatment completion, the effect of TVT was maintained, both in pain and ADLs. Proprioceptors, such as muscle spindles, exhibit specific responsive vibration frequency. Vibratory stimulation that matches the frequency of these proprioceptors activates afferent nerve fibers, thereby enhancing proprioceptive functions [7]. Given the reduced responsiveness to vibratory stimuli in impaired proprioceptors, the tendency of illusory movement is less likely to occur, resulting in impaired proprioception. Sensory-motor incongruity has led to a new concept in proprioceptive pain. The hypothesis proposed by Harris in 1999 suggested that discordance between awareness of motor intention and muscle or joint proprioception may lead to pathologic chronic pain [29]. Limited proprioceptive feedback leads to pathological pain in various parts of the body. Therapies that restore the integrity of cortical information processing for pain without underlying pathological causes have the potential to replace conventional analgesic treatments. Our findings indicate an increase in the biological response to vibratory stimuli, specifically proprioception, was enhanced through the continuous application of the corresponding vibration specific to the impaired proprioceptors. The significant improvements in

proprioceptive function, pain intensity, and ADLs associated with treatment, and their tendency to worsen after the end of the procedure, provided sufficient evidence that the effect of TVT on proprioception contributes to symptom improvement in older patients with chronic LBP.

The greatest concern regarding adverse reactions to TVT pertains to its effects on normal proprioceptors. Vibratory stimulation targeting dysfunctional receptors did not worsen the perturbation and effectively enhanced the responsiveness of the dysfunctional receptors. This indicates that the treatment affected the targeted proprioceptors without adversely affecting the overall center of gravity sway. Before this study, our research group evaluated the effects of therapeutic vibrations on proprioceptors. Vibratory stimulation with frequencies corresponding to the dysfunctional proprioceptors immediately improved their proprioceptive function. Simultaneously, the response of non-targeted receptors to vibratory stimulation was reduced, and proprioception worsened [30]. Lasting hyperactivation of proprioceptors continuously stimulates the reflex arc in the spinal cord, further inducing microglial activation, leading to the initiation and maintenance of pain [31]. Given that vibratory stimulation can cause harm to humans and induce LBP [30], we emphasize the importance of functional proprioception diagnosis in the development of vibration-based pain therapy. Vibratory therapy targeting dysfunctional proprioceptors is ideal for treating deteriorating proprioceptive function.

There have been several reports of whole-body vibration (WBV) as a mechanism by which vibration stimulation of skeletal muscles contributes to pain reduction. Vibratory stimulation has been suggested to increase muscle flexibility by activating the Ia inhibitory interneurons of the antagonist muscle following vibration [32]. Additionally, it is proposed that vibration may induce changes in intramuscular coordination, thereby reducing the braking force around the joints in the lower back [33]. One effect on proprioception is the activation of inhibitory circuits in the dorsal horn of the spinal cord, facilitated by the transmission from proprioceptive receptors through sensory fibers type Aβ. These circuits subsequently diminish the pain input to C fibers, resulting in pain reduction [34]. Moreover, increased secretion of myokine, an anti-inflammatory cytokine, has been reported as a response of skeletal muscles to WBV [35], and it may be expected that the contraction of skeletal muscles may induce analgesia through exercise. While the mechanism behind the pain-relieving effect of vibration stimulation on skeletal muscles remains inconclusive, it is important to note that the vibration stimulation used in this study was not WBV; however, the vibration device used in this study was operated in accordance with the guidelines on WBV [20]. The fact that TVT was effective in improving proprioceptive function and LBP at the same time in the present study supports the relationship between proprioception and LBP.

The greatest strength of this study lies in its rigorous diagnosis of proprioceptive dysfunction and targeted identification of impaired proprioceptors, which allowed for the precise application of vibratory stimulation without adverse effects. This TVT effectively improved proprioception and alleviated chronic LBP in older patients with impaired proprioceptive function. With the potential for sustained effects through treatment period modifications, TVT represents a promising modality for chronic pain management.

However, the study has some limitations. First, using a single arm without a control group obscured the interpretation of data integrity and validity regarding the treatment effects obtained. Establishing a placebo control group requires applying non-therapeutic vibratory stimulation in patients with chronic LBP and impaired proprioceptive function. However, as aforementioned, vibratory stimulation targeting proprioceptors other than dysfunctional ones should be avoided, as it could potentially exacerbate the deterioration in proprioceptive function. To address these limitations, we meticulously assessed the proprioceptive function of patients with chronic LBP and developed a treatment protocol that exclusively targeted the

receptors with reduced function. This approach relies on assessments conducted at the site of proprioceptive dysfunction and the corresponding frequency band. Therefore, the effectiveness of TVT was examined by evaluating whether the improvement in proprioceptive function could be obtained consistently with beneficial changes in LBP. Second, there is controversy surrounding the appropriateness of the treatment duration setting. Although the patients demonstrated significant improvements in LBP and proprioceptive function after 2 weeks of treatment, a worsening trend was observed at 2 weeks following the treatment completion. A comparison of different treatment periods and studies with a longer follow-up period are needed to determine the appropriate treatment duration.

## Conclusion

TVT, involving vibratory stimulation applied to impaired proprioceptors at a frequency corresponding to the receptor, improves proprioception and LBP in older patients with impaired proprioceptive function. Further sustained effects could be anticipated through modification of the treatment period. This therapy presents a potential treatment modality for chronic pain.

## Supporting information

**S1 Checklist. TREND statement checklist.**
(PDF)

**S1 File. Trial study protocol (English translation).**
(PDF)

**S2 File. Trial study protocol (Original).**
(PDF)

## Acknowledgments

We are grateful to all the patients who agreed to participate in the study, the staff who delivered the intervention, and the surgeons who helped with patient recruitment. We thank the Nagoya Institute of Clinical Pharmacology for providing an independent safety monitoring board and managing the data. We also appreciate the administrative assistance of Yayoi Sato, Junk Suzuki, and Miki Morita.

## Author Contributions

**Conceptualization:** Yoshihito Sakai.

**Data curation:** Keitaro Kawai, Jo Fukuhara, Kazunori Yamazaki.

**Formal analysis:** Tadashi Ito.

**Investigation:** Yoshifumi Morita, Keitaro Kawai, Jo Fukuhara, Norimitsu Wakao, Hiroki Matsui.

**Methodology:** Yoshihito Sakai.

**Supervision:** Yoshifumi Morita.

**Validation:** Yoshifumi Morita, Tsuyoshi Watanabe.

**Writing – original draft:** Yoshihito Sakai.

**Writing – review & editing:** Yoshifumi Morita, Tadashi Ito, Tsuyoshi Watanabe, Norimitsu Wakao, Hiroki Matsui.

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
