## [Decision Letter · Decision Letter 0]

20 Mar 2024

PONE-D-23-38713Targeted vibratory therapy as a treatment for proprioceptive dysfunction: clinical trial in older patients with chronic low back painPLOS ONE

Dear Dr. Sakai,

Thank you for submitting your manuscript to PLOS ONE. After careful consideration, we feel that it has merit but does not fully meet PLOS ONE’s publication criteria as it currently stands. Therefore, we invite you to submit a revised version of the manuscript that addresses the points raised during the review process.

We look forward to receiving your revised manuscript.

Kind regards,

Jose María Blasco, Ph.D.

Academic Editor

PLOS ONE

Journal Requirements:

We are grateful to all the patients who agreed to participate in the study, the staff who delivered the intervention, and the surgeons who helped with patient recruitment. We thank the Nagoya Institute of Clinical Pharmacology for providing an independent safety monitoring board and managing the data. We also appreciate the administrative assistance of Yayoi Sato, Junk Suzuki, and Miki Morita. 

Financial and Material Support: This work was supported by National Center for Geriatrics and Gerontology (21-32). The funders were not involved in the design or conduct of the study, in the collection, management, analysis or interpretation of the data, or in the preparation, review or approval of the manuscript.

4. Please amend the manuscript submission data (via Edit Submission) to include author Dr. Kazunori Yamazaki.

Reviewers' comments:

Reviewer's Responses to Questions

**Comments to the Author**

1. Is the manuscript technically sound, and do the data support the conclusions?

Reviewer #1: Yes

Reviewer #2: Yes

Reviewer #3: Yes

2. Has the statistical analysis been performed appropriately and rigorously? 

Reviewer #1: Yes

Reviewer #2: Yes

Reviewer #3: Yes

3. Have the authors made all data underlying the findings in their manuscript fully available?

Reviewer #1: Yes

Reviewer #2: Yes

Reviewer #3: Yes

4. Is the manuscript presented in an intelligible fashion and written in standard English?

Reviewer #1: Yes

Reviewer #2: Yes

Reviewer #3: Yes

5. Review Comments to the Author

Reviewer #1: Major Revision Required

This is a very interesting feasibility study although the presentation needs improvement.

This paper has two distinct sections. One compares the characteristics of patients with and without proprioceptive dysfunction (see Table 1). The second tests the outcome in those with dysfunction following targeted vibratory therapy (see Table 2, 3 & 4). From the title of the paper, the latter is clearly the major concern. The distinction between these two components needs to be reflected in the presentation – particularly the Abstract.

Although, the changes suggested below are not major, they will require a substantial revision of the abstract, sample size, statistical analysis, and tabular presentations

Some other points are

Page Lines

3, 4 Abstract This does not really describe the study design and results in sufficient detail. Requires a major restructuring.

13 213-7 This section needs rephrasing. I suggest:

‘The aim was to demonstrate the same clinical difference in NRS scores as those achieved following 2-week opioid therapy for older patients with chronic LBP [20]. This suggested a difference in NRS of 1·7 points (SD = 3.1). Then for a paired t-test, at a two-sided significance level of 0.05 and 80% power, requires 29 patients to be recruited.’

219-220 This does not describe how the statistical comparisons of Tables 2, 3 & 4 were conducted. The sample size calculation depends on the use of the Student’s t-test so this needs to be mentioned here. Also CIs are used in the tables but there is no mention here.

13 226 More useful to give the minimum, median and maximum ages concerned.

15 Table 1 This table is far too detailed. It would help if the number of decimal places used was reduced. For example, for total focus length (eye open) replace ‘415.46 (125.28)’ by ‘416 (125)’.

Not clear what is meant by ‘RMS’?

Table 1 & elsewhere Reduce p-values to 2 significant figures. For example,’ 0.7251 to 0.73’ and ‘0.0919 to 0.092’.

17-19 Tables 2, 3 and 4 Suggest that the current column 3, precedes Column 2 and that the current Column 7 follows the current Column 5. Thus the current Column 6 then becomes Column 7.

Suggest the order within current headings of Columns 5, 6, 7 should be ‘difference, 99%CI, then p-values’ as the magnitude of the differences is the principal focus.

Needs to indicate how many patients are concerned with each variable – is it always 32?

Not sure if Column 6 is worth reporting although the magnitude of that difference between weeks 2 and 4 weeks deserves comment. Do they indicate that (more-or-less) the value of the target therapy plateau’s after week 2? In any event, I suggest the statistical test results are omitted.

Reviewer #2: Congratulations. The subject of this study is highly relevant. Moreover, important findings are presented. However, the presentation of the study must be improved. Please, find my comments and suggestions in the attached file. I am suggesting an inclusion of a reference, that is the guidelines about the utilizations of mechanical vibrations in health sciences.

Reviewer #3: Title: Targeted vibratory therapy as a treatment for proprioceptive dysfunction: clinical trial in older patients with chronic low back pain

Journal: PLOS ONE

Manuscript ID PONE-D-23-38713

The current study aimed to investigate the feasibility of improving proprioceptive function and its effect on alleviating chronic LBP in older patients through targeted vibratory therapy (TVT) administration. The approach is original. The manuscript reads smoothly and is easy to understand. The aims, scope, and results of the study are clearly stated. I have very much enjoyed reading this paper. I find it interesting and clearly written and satisfying also all the other publication criteria of the “PLOS ONE”. The study provides a very valuable addition to this line of research and adds relevantly to the subject with additional original findings. I thus find that this paper definitively delivers results that will surely be of interest to the readership of the “PLOS ONE”. I recommend the publication of this paper after revision. The authors must develop the limitations of the study. The authors must use REF from serious journals and indexed. I recommend the addition of the following references that will increase the methodology and discussion sections that appears still poor. They are the guideline of this area of this research and must be used like a reference of methodology.

• Wuestefeld A, FuermaierABM, Bernardo-Filho M, da Cunhade Sá-CaputoD, Rittweger J, Schoenau E, et al. (2020). Towards reporting guidelines of research using whole-body vibration as training or treatment regimen in human subjects—A Delphi consensus study. PLoSONE, 15(7):e0235905 https://doi.org/10.1371/journal.pone.0235905

• Acute Effects of Whole-Body Vibration on the Pain Level, Flexibility, and Cardiovascular Responses in Individuals With Metabolic Syndrome. Dose-Response: October-December 2018:1-9. https://doi.org/10.1177/1559325818802139.

• Relevance of Whole-Body Vibration Exercises on Muscle Strength/Power and Bone of Elderly Individuals. DOI https://doi.org/10.1177/1559325818813066

• Whole-body vibration improves the functional parameters of individuals with metabolic syndrome: An exploratory study. DOI https://doi.org/10.1186/s12902-018-0329-0

• Do whole body vibration exercises affect lower limbs neuromuscular activity in populations with a medical condition? A systematic review. DOI https://doi.org/10.3233/RNN-170765

• Whole-body vibration improves the functional parameters of individuals with metabolic syndrome: An exploratory study. https://doi.org/10.1186/s12902-018-0329-0

• Potential application of whole body vibration exercise for improving the clinical conditions of covid-19 infected individuals: A narrative review from the world association of vibration exercise experts (wavex) panel DOI 10.3390/ijerph17103650

• Whole-body vibration improves the functional parameters of individuals with metabolic syndrome: An exploratory study 10.1186/s12902-018-0329-0

• Attitudes to knee osteoarthritis and total knee replacement in Arab women: A qualitative study 10.1186/1756-0500-6-406

• Moreira-Marconi, E. et al. Evaluation of the temperature of posterior lower limbs skin during the whole body vibration measured by infrared thermography: Cross-sectional study analysis using linear mixed effect model (2019) PLoS ONE, 14 (3), art. no. e0212512,

6. PLOS authors have the option to publish the peer review history of their article (what does this mean?). If published, this will include your full peer review and any attached files.

Reviewer #1: No

Reviewer #2: **Yes: **Mario Bernardo-Filho

Reviewer #3: No

---

## [Author Response · Author response to Decision Letter 0]

8 May 2024

AUTHORS’ RESPONSES TO REVIEWERS’ COMMENTS

We would like to thank the reviewers for their critique on our study, entitled “Targeted vibratory therapy as a treatment for proprioceptive dysfunction: clinical trial in older patients with chronic low back pain.” Their comments have helped us improve the quality of our work. The point-by-point responses to their comments are presented below. The revisions in our manuscript are highlighted in gray.

Reviewer 1

Abstract This does not really describe the study design and results in sufficient detail. Requires a major restructuring.

Response: We would like to thank the reviewer for evaluating our manuscript and for the insightful comments. In the Abstract, we have clarified the study design and added a comparison of proprioceptive dysfunction and normal function to clarify the two components of this study, as per the reviewer’s suggestion. The revised Abstract is as follows:

“Proprioceptive function declines with age, leading to falls, pain, and difficulties in performing activities of daily living among older adults. Although individuals with low back pain (LBP) exhibit decreased lumbosacral proprioception in various postures, the mechanism by which reduced proprioceptive function causes LBP remains uncertain. Vibratory stimulation may enhance proprioceptive function; however, its efficacy as a treatment for LBP has not been investigated. In this study, we investigated the feasibility of improving proprioceptive function and its effect on alleviating chronic LBP in older patients through targeted vibratory therapy (TVT) administration. Older patients aged >65 years with non-specific chronic LBP were recruited for a single-arm designed trial. TVT involved applying vibratory stimulation, matching the frequency of dysfunctional receptors, for 1 min daily over 14 days to activate proprioceptors, with patients performing TVT three times daily at home. In cases with reduced proprioceptive function at multiple sites, the targeted treatment was aimed at the lowest frequency band value. Biological responses to vibratory stimulation targeting reduced proprioception in the trunk or lower legs were measured at sites with impaired function. Overall, 56 patients with chronic LBP were enrolled, of whom 32 were recruited for treatment based on a proprioceptive dysfunction diagnosis, compared to 24 patients diagnosed as normal, with no significant differences observed in factors related to sarcopenia, and no clinical features in proprioception. No adverse events were observed in any patient. After 2 weeks of TVT, the numerical pain rating scale score improved to <3 points in 78.1% of patients, with 73.1% achieving a score ≤ 3 points. Proprioceptive function improved in 81.3% of cases, and engagement in activities of daily living improved significantly. TVT, involving vibratory stimulation at a receptor-specific frequency, demonstrated efficacy in improving proprioception and alleviating LBP in older patients with impaired proprioceptive function without affecting non-targeted proprioceptors.”

P213 line 213-217 This section needs rephrasing.

Response: We have made the following revisions in accordance with the reviewer’s suggestion.

“The aim was to demonstrate the same clinical difference in the NRS scores as those achieved following 2-week opioid therapy for older patients with chronic LBP [22]. This suggested a difference in the NRS of 1.7 (SD, 3.1) points. For a paired t-test with a two-sided significance level of 0.05 and 80% power, requires 29 patients needed to be recruited.” (Lines 222-225)

P213 line 219-220 This does not describe how the statistical comparisons of Tables 2, 3 & 4 were conducted. The sample size calculation depends on the use of the Student’s t-test so this needs to be mentioned here. Also, CIs are used in the tables but there is no mention here.

Response: In Table 2, the Mann–Whitney U test was used for the two-group comparison in the presence of proprioceptive dysfunction, and a description of CI was added. The following changes were made in the “Data Management and Statistical Analysis” subsection:

“The aim was to demonstrate the same clinical difference in the NRS scores as those achieved following 2-week opioid therapy for older patients with chronic LBP [22]. This suggested a difference in the NRS of 1.7 (SD, 3.1) points. For a paired t-test with a two-sided significance level of 0.05 and 80% power, requires 29 patients needed to be recruited.

We compared the baseline characteristics of patients with and without proprioceptive dysfunction using the Mann–Whitney U test. Improvements in LBP, activities of daily living (ADLs), and proprioceptive function were assessed by measuring changes before and after treatment. The statistical significance of the differences in pain intensity, ADLs, and proprioceptive function was compared between pre-treatment and 2 weeks after treatment, 2 weeks and 4 weeks after treatment, and pre-treatment and 4 weeks after treatment using the Wilcoxon signed-rank test. The results are presented as mean differences for continuous outcomes with 99% confidence intervals (CIs).” (Lines 222-232)

Page13 line 226 More useful to give the minimum, median and maximum ages concerned.

Response: Please note that the minimum, median, and maximum ages were added. The revised part is as follows:

“Ultimately, 56 patients with chronic LBP (mean age, 76.38±5.30 years; minimum, 67 years; median, 76 years; maximum, 87 years; sex, 28 male and 28 female individuals) were enrolled, and all patients underwent proprioceptive function diagnostic evaluation.” (Lines 236-238)

Page 15 Table 1 This table is far too detailed. It would help if the number of decimal places used was reduced. For example, for total focus length (eye open) replace ‘415.46 (125.28)’ by ‘416 (125)’.

Response: In accordance with the reviewer's suggestion, the number of digits in Table 1 was reduced to zero by rounding off the decimal point in values with more than three digits.

Not clear what is meant by ‘RMS’?

Response: A description of RMS can be found in line157 of the revised manuscript (Methods section). For clarity, we have also added the following sentence to table footnotes.

“The root mean square (RMS) was calculated by the difference in CoP data between the pre- and vib- sections. (See Diagnosis of proprioception).”

Table 1 & elsewhere Reduce p-values to 2 significant figures. For example,’ 0.7251 to 0.73’ and ‘0.0919 to 0.092’.

Response: We have made the appropriate revisions in Table 1-4.

Suggest the order within current headings of Columns 5, 6, 7 should be ‘difference, 99%CI, then p-values’ as the magnitude of the differences is the principal focus.

Response: We agree with this comment, and have corrected the order of Columns 5, 6, and 7 in all of Table 2, 3, and 4 from ↑ to difference, 99%CI, and p value.

Needs to indicate how many patients are concerned with each variable – is it always 32?

Response: The number of patients for each variable is always 32, as all 32 patients treated were evaluated. No patient dropped out during the study.

Not sure if Column 6 is worth reporting although the magnitude of that difference between weeks 2 and 4 weeks deserves comment. Do they indicate that (more-or-less) the value of the target therapy plateau’s after week 2? In any event, I suggest the statistical test results are omitted.

Response: We would like to thank the reviewer for the valuable suggestions. Following a 2-week treatment period, we have undertaken the assessment of treatment persistence by temporarily halting TVT and subsequently evaluating the patients’ condition after a further 2 weeks. Thus, we opted to describe the results of the study, as we believe that demonstrating that patients maintained improved condition post-treatment, despite experiencing slight deterioration due to the treatment interruption, could serve as justification for a reasonable TVT treatment period. The study protocol also calls for an evaluation at 2 weeks after the end of treatment, and we think that it is desirable to include this information without omitting it. We appreciate the reviewer’s comments, and we will use them as a reference for future protocol development.

 

Reviewer 2

Please, find my comments and suggestions in the attached file. 

P6 line81 I suggest adding references here. Some suggestions are:

1: van Heuvelen MJG, Rittweger J, Judex S, Sañudo B, Seixas A, Fuermaier ABM, Tucha O, Nyakas C, Marín PJ, Taiar R, Stark C, Schoenau E, Sá-Caputo DC, Bernardo-Filho M, van der Zee EA. Reporting Guidelines for Whole-Body Vibration Studies in Humans, Animals and Cell Cultures: A Consensus Statement from an International Group of Experts. Biology (Basel). 2021, 27;10(10):965. doi: 10.3390/biology10100965. 

2: Sá-Caputo D, Taiar R, Martins-Anjos E, Seixas A, Sartório A, Sanudo B, Sonza A, Amaral V, Lacerda A, Gomes-Neto M, Moura-Filho O, Oliveira L, Bachur J, Bernardo-Filho M. Does the mechano-biomodulation vibration lead to biological responses on human beings. Series on Biomechanics, 2023, 37 (2): 3-17. doi: 10.7546/SB.01.02.2023

I am suggesting an inclusion of a reference, that is the guidelines about the utilizations of mechanical vibrations in health sciences.

Response: We would like to thank the reviewer for evaluating our manuscript and for the insight comments. Although this study is not a report on WBV treatment, we believe that it should be conducted in accordance with these guidelines in terms of vibration to the human body. We have published the information on the vibration used in this study in thesubsection "TVT for proprioceptive dysfunction" in the Methods section, according to the suggested Reference #1. The added part is as follows:

“The vibratory intervention programs of this trial are structured according to previously published guidelines for whole-body vibration studies in information about the device and vibration [20]. The device used in this therapeutic trial is non-commercial (self-built), and uses the same vibrators as the diagnostic devise described above. Briefly, the device is 60 mm * 75 mm * 3 mm large, has a mass of 52.3 g, and can deliver sinusoidal (deviation <5%) vibration per vibrator. The device can produce vibrations with peak-to-peak displacement up to 1.6 mm within a frequency range of 30–53Hz, 56–00Hz, and 140–250Hz. The vibrator produced sinusoidal accelerations primarily in the vertical component, with minimal variation in acceleration direction throughout each sinusoidal cycle.” (Lines 183-191)

P7 line 102 Defne it.

Response: Please note that the name of the facility was added.

“a single institute (National Center for Geriatrics and Gerontology).”

P7 line 106 add reference

Response: Please note that we have added a relevant references.

Reference

Shafshak TS, Elnemr R. The visual analogue scale versus numerical rating scale in measuring pain severity and predicting disability in low back pain. J Clin Rheumatol. 2021;27: 282-285. doi: 10.1097/RHU.0000000000001320.

P19 line 223 Put all "p" related to statistical analysis in italic format, here abd throughout the manuscript.

Response: Please note that we have used italics when presenting the p-values.

P23 line 377-379 I suggest considering these sentences as the strengths of the study. Use a single paragraph with these statements.

Response: We would like to thank the reviewer for the suggestion. Please note that we have moved the sentence from Lines 377–380 before the limitation and used a single paragraph to describe the strengths of this study. The added part is as follows:

“The greatest strength of this study lies in its rigorous diagnosis of proprioceptive dysfunction and targeted identification of impaired proprioceptors, which allowed for the precise application of vibratory stimulation without adverse effects. This TVT effectively improved proprioception and alleviated chronic LBP in older patients with impaired proprioceptive function. With the potential for sustained effects through treatment period modifications, TVT represents a promising modality for chronic pain management”. (Lines 403-408)

Reviewer 3

The current study aimed to investigate the feasibility of improving proprioceptive function and its effect on alleviating chronic LBP in older patients through targeted vibratory therapy (TVT) administration. The approach is original. The manuscript reads smoothly and is easy to understand. The aims, scope, and results of the study are clearly stated. I have very much enjoyed reading this paper. I find it interesting and clearly written and satisfying also all the other publication criteria of the “PLOS ONE”. The study provides a very valuable addition to this line of research and adds relevantly to the subject with additional original findings. I thus find that this paper definitively delivers results that will surely be of interest to the readership of the “PLOS ONE”. I recommend the publication of this paper after revision. The authors must develop the limitations of the study. The authors must use REF from serious journals and indexed. I recommend the addition of the following references that will increase the methodology and discussion sections that appears still poor. They are the guideline of this area of this research and must be used like a reference of methodology.

• Wuestefeld A, FuermaierABM, Bernardo-Filho M, da Cunhade Sá-CaputoD, Rittweger J, Schoenau E, et al. (2020). Towards reporting guidelines of research using whole-body vibration as training or treatment regimen in human subjects—A Delphi consensus study. PLoSONE, 15(7):e0235905 https://doi.org/10.1371/journal.pone.0235905

• Acute Effects of Whole-Body Vibration on the Pain Level, Flexibility, and Cardiovascular Responses in Individuals With Metabolic Syndrome. Dose-Response: October-December 2018:1-9. https://doi.org/10.1177/1559325818802139.

• Relevance of Whole-Body Vibration Exercises on Muscle Strength/Power and Bone of Elderly Individuals. DOI https://doi.org/10.1177/1559325818813066

• Whole-body vibration improves the functional parameters of individuals with metabolic syndrome: An exploratory study. DOI https://doi.org/10.1186/s12902-018-0329-0

• Do whole body vibration exercises affect lower limbs neuromuscular activity in populations with a medical condition? A systematic review. DOI https://doi.org/10.3233/RNN-170765

• Whole-body vibration improves the functional parameters of individuals with metabolic syndrome: An exploratory study. https://doi.org/10.1186/s12902-018-0329-0

• Potential application of whole body vibration exercise for improving the clinical conditions of covid-19 infected individuals: A narrative review from the world association of vibration exercise experts (wavex) panel DOI 10.3390/ijerph17103650

• Whole-body vibration improves the functional parameters of individuals with metabolic syndrome: An exploratory study 10.1186/s12902-018-0329-0

• Attitudes to knee osteoarthritis and total knee replacement in Arab women: A qualitative study 10.1186/1756-0500-6-406

• Moreira-Marconi, E. et al. Evaluation of the temperature of posterior lower limbs skin during the whole body vibration measured by infrared thermography: Cross-sectional study analysis using linear mixed effect model (2019) PLoS ONE, 14 (3), art. no. e0212512

Response: We would like to thank the reviewer for evaluating our manuscript and for the insightful comments. Please note that we have added the following statement regarding vibration as a treatment in Methods section, as the WBV guidelines are needed to clarify the information on vibration. The following references were used for the guideline, as they are more recent.

van Heuvelen MJG, Rittweger J, Judex S, Sañudo B, Seixas A, Fuermaier ABM, Tucha O, Nyakas C, Marín PJ, Taiar R, Stark C, Schoenau E, Sá-Caputo DC, Bernardo-Filho M, van der Zee EA. Reporting Guidelines for Whole-Body Vibration Studies in Humans, Animals and Cell Cultures: A Consensus Statement from an International Group of Experts. Biology (Basel). 2021, 27;10(10):965. doi: 10.3390/biology10100965. 

The added part is as follows:

“The vibratory intervention programs of this trial are structured according to previously published guidelines for whole-body vibration studie

---

## [Decision Letter · Decision Letter 1]

28 May 2024

PONE-D-23-38713R1Targeted vibratory therapy as a treatment for proprioceptive dysfunction: clinical trial in older patients with chronic low back painPLOS ONE

Dear Dr. Sakai,

Thank you for submitting your manuscript to PLOS ONE. After careful consideration, we feel that it has merit but does not fully meet PLOS ONE’s publication criteria as it currently stands. Therefore, we invite you to submit a revised version of the manuscript that addresses the points raised during the review process.

We look forward to receiving your revised manuscript.

Kind regards,

Jose María Blasco, Ph.D.

Academic Editor

PLOS ONE

Journal Requirements:

Reviewers' comments:

Reviewer's Responses to Questions

**Comments to the Author**

1. If the authors have adequately addressed your comments raised in a previous round of review and you feel that this manuscript is now acceptable for publication, you may indicate that here to bypass the “Comments to the Author” section, enter your conflict of interest statement in the “Confidential to Editor” section, and submit your "Accept" recommendation.

Reviewer #1: All comments have been addressed

Reviewer #3: All comments have been addressed

Reviewer #4: (No Response)

2. Is the manuscript technically sound, and do the data support the conclusions?

Reviewer #1: (No Response)

Reviewer #3: Yes

Reviewer #4: Yes

3. Has the statistical analysis been performed appropriately and rigorously? 

Reviewer #1: (No Response)

Reviewer #3: Yes

Reviewer #4: Yes

4. Have the authors made all data underlying the findings in their manuscript fully available?

Reviewer #1: (No Response)

Reviewer #3: Yes

Reviewer #4: Yes

5. Is the manuscript presented in an intelligible fashion and written in standard English?

Reviewer #1: (No Response)

Reviewer #3: Yes

Reviewer #4: Yes

6. Review Comments to the Author

Reviewer #1: Accept

The authors have made substantial changes to the paper which, I hope, will make it easier for a clinical reader to digest. I have no further comments.

Reviewer #3: no more comments. The authors replayed to my request. The paper is interesting and can be published in the present form

Reviewer #4: First of all, I want to congratulate you for the work you have done. I communicate a series of suggestions that could improve the quality of your work:

- Introduction: Although it is not necessary to have a structured abstract, we recommend rewriting it in order to highlight and make each of the sections clear. I also recommend reviewing the writing as it can be improved.

- Line 47-48: Explain this sentence, it’s not understood.

- Line 63: Fixe double spacing.

- Line 100: Explain if participants were asked to sign the informed consent and if it was explained to them.

- Line 289, 302 and 317: Please, explain under the table what ‘treatment+’ and ‘treatment-‘ mean.

7. PLOS authors have the option to publish the peer review history of their article (what does this mean?). If published, this will include your full peer review and any attached files.

Reviewer #1: No

Reviewer #3: No

Reviewer #4: No

---

## [Author Response · Author response to Decision Letter 1]

4 Jun 2024

We appreciate the reviewers for the critique on our study titled; “Targeted vibratory therapy as a treatment for proprioceptive dysfunction: clinical trial in older patients with chronic low back pain.” . We have revised the manuscript according to the reviewer’s comments. Revisions are highlighted in gray in the attached manuscript.

Reviewer #4: First of all, I want to congratulate you for the work you have done. I communicate a series of suggestions that could improve the quality of your work:

- Introduction: Although it is not necessary to have a structured abstract, we recommend rewriting it in order to highlight and make each of the sections clear. I also recommend reviewing the writing as it can be improved.

Answer: Thank you for the suggestion. Accordingly, the abstract has been structured to clearly demarcated each of the sections and the language of the abstract has been rechecked.

- Line 47-48: Explain this sentence, it’s not understood.

Answer: Apologies for the lack of clarity before. Lines 47-49 “Biological responses to vibratory stimulation targeting reduced proprioception in the trunk or lower legs were measured at sites with impaired function.” have now been revised as follows:

LBP and proprioceptive function were evaluated at 2 weeks after TVT and at 2 weeks after the end of TVT in patients with declined proprioception in the trunk or lower extremities.

- Line 63: Fixe double spacing.

Answer: Thank you for pointing this out. The manuscript has been double-spaced throughout.

- Line 100: Explain if participants were asked to sign the informed consent and if it was explained to them.

Answer: Thank you for this advice. The following text has been added.

Participants were briefed on the study and each of them provided informed consent.

- Line 289, 302 and 317: Please, explain under the table what ‘treatment+’ and ‘treatment-‘ mean.

Answer: Apologies for the lack of clarity before. The following explanatory text has been added.

treatment+ indicates 2 weeks after treatment and treatment- indicates 2 weeks after the end of treatment.

---

## [Decision Letter · Decision Letter 2]

25 Jun 2024

Targeted vibratory therapy as a treatment for proprioceptive dysfunction: clinical trial in older patients with chronic low back pain

PONE-D-23-38713R2

Dear Dr. Sakai,

We’re pleased to inform you that your manuscript has been judged scientifically suitable for publication and will be formally accepted for publication once it meets all outstanding technical requirements.

Kind regards,

Jose María Blasco, Ph.D.

Academic Editor

PLOS ONE

**Comments to the Author**

1. If the authors have adequately addressed your comments raised in a previous round of review and you feel that this manuscript is now acceptable for publication, you may indicate that here to bypass the “Comments to the Author” section, enter your conflict of interest statement in the “Confidential to Editor” section, and submit your "Accept" recommendation.

Reviewer #4: All comments have been addressed

2. Is the manuscript technically sound, and do the data support the conclusions?

Reviewer #4: Yes

3. Has the statistical analysis been performed appropriately and rigorously? 

Reviewer #4: Yes

4. Have the authors made all data underlying the findings in their manuscript fully available?

Reviewer #4: Yes

5. Is the manuscript presented in an intelligible fashion and written in standard English?

Reviewer #4: Yes

6. Review Comments to the Author

Reviewer #4: Thank you very much for having attended to my requests. For my part, there is no need to make any more changes.

7. PLOS authors have the option to publish the peer review history of their article (what does this mean?). If published, this will include your full peer review and any attached files.

Reviewer #4: No

---

## [Editor Report · Acceptance letter]

10 Jul 2024

PONE-D-23-38713R2 

PLOS ONE

Dear Dr. Sakai, 

I'm pleased to inform you that your manuscript has been deemed suitable for publication in PLOS ONE. Congratulations! Your manuscript is now being handed over to our production team.

Kind regards, 

on behalf of

Dr. Jose María Blasco 

Academic Editor

PLOS ONE